# Unraveling *TNXB* Epigenetic Alterations Through Genome-Wide DNA Methylation Analysis and Their Implications for Colorectal Cancer

**DOI:** 10.3390/ijms26157197

**Published:** 2025-07-25

**Authors:** Jesús Pilo, Alejandro Rego-Calvo, Libia-Alejandra García-Flores, Isabel Arranz-Salas, Ana Isabel Alvarez-Mancha, Andrea G. Izquierdo, Ana B. Crujeiras, Julia Alcaide, Maria Ortega-Castan, Hatim Boughanem, Manuel Macías-González

**Affiliations:** 1Department of Endocrinology and Nutrition, Virgen de la Victoria University Hospital, 29010 Malaga, Spain; jesuspilor@gmail.com (J.P.); alejandro.rego.calvo@gmail.com (A.R.-C.); lic.libiaalejandra@gmail.com (L.-A.G.-F.); 2Institute of Biomedical Research in Malaga (IBIMA)-Bionand Platform, University of Malaga, 29010 Malaga, Spain; 3Spanish Biomedical Research Center in Physiopathology of Obesity and Nutrition (CIBERObn), Instituto de Salud Carlos III, 28029 Madrid, Spain; andrea.gonzalez.izquierdo@hotmail.com (A.G.I.); acrujeirasm@gmail.com (A.B.C.); 4Unidad de Gestión Clínica (UGC) de Anatomía Patológica, Instituto de Investigación Biomédica de Málaga (IBIMA), Hospital Universitario Virgen de la Victoria, 29010 Malaga, Spain; isabellanz@yahoo.es (I.A.-S.); aiamisa@gmail.com (A.I.A.-M.); 5Epigenomics in Endocrinology and Nutrition Group, Epigenomics Unit, Instituto de Investigacion Sanitaria de Santiago de Compostela (IDIS), Complejo Hospitalario Universitario de Santiago de Compostela (CHUS/SERGAS), 15706 Santiago de Compostela, Spain; 6Servicio de Oncología Médica, Hospital Regional Universitario de Málaga, Instituto de Investigación Biomédica de Málaga (IBIMA), 29010 Malaga, Spain; julia.alcaide.sspa@juntadeandalucia.es; 7Unidad de Gestion Clinica Cirugía General y del Aparato Digestivo, Virgen de la Victoria University, 29010 Malaga, Spain; maorca@hotmail.es; 8Unidad de Gestión Clinica Medicina Interna, Lipids and Atherosclerosis Unit, Maimonides Institute for Biomedical Research in Córdoba, Reina Sofia University Hospital, University of Córdoba, 14004 Cordoba, Spain

**Keywords:** DNA methylation, TNXB, colorectal cancer, 450K, epigenetic

## Abstract

Aberrant DNA methylation has been shown to be a fingerprint characteristic in human colorectal tumors. In this study, we hypothesize that investigating global DNA methylation could offer potential candidates for clinical application in CRC. The epigenome-wide association analysis was conducted in both the tumor area (N = 27) and the adjacent tumor-free (NAT) area (N = 15). We found 78,935 differentially methylated CpG sites (DMCs) (FDR < 0.05), 42,888 hypomethylated and 36,047 hypermethylation showing overall hypomethylation. Gene ontology and KEGG analysis of differentially methylated genes showed significant enrichment in developmental genes, as well as in genes involved in metabolic processes and the cell cycle, such as the TFGβ and cAMP signaling pathways. Through filtered analysis, we identified *TNXB* as the most epigenetically dysregulated gene, hypomethylated and downregulated in CRC (both with *p* < 0.001) and associated with poor overall survival. In the functional analysis, *TNXB* was epigenetically regulated in a dose-dependent manner, suggesting a potential role in CRC. The epigenetic dysregulation and functional role of *TNXB* in CRC could have clinical implications, serving as indicators of malignant potential, with adverse effects associated with disease origin and progression in CRC.

## 1. Introduction

According to the data from the International Agency for Research on Cancer (IARC) and the Global Observatory for Cancer (GLOBOCAN), colorectal cancer (CRC) ranks as the third most commonly diagnosed cancer, constituting 10% of all cancer diagnoses globally, and the second leading cause of cancer-related deaths [1]. Despite increased survival rates in metastatic cancer cases, mostly due to improved screening, the overall survival rate for advanced cases remains low [2]. In fact, epigenetic modifications constitute a prevalent occurrence in all human cancers. Epigenetics refers to heritable changes in gene expression without alterations in the DNA sequence. These modifications mainly include DNA methylation and histone modifications, which regulate gene accessibility. In CRC, epigenetic alterations, such as aberrant DNA methylation, play a key role in tumor initiation and progression, notably marked by aberrant DNA methylation in human colorectal tumors [3], and commonly exhibiting global genome hypomethylation [4,5]. This hypomethylation potentially plays a critical role in promoting carcinogenesis by modulating genomic instability, poorer prognoses and response to chemotherapy [6]. Alterations in DNA methylation have been shown to be useful for clinical application, since they may occur early in oncogenesis, are stable, reversible, and can be assayed in many tissues by non-invasive methods [7].

Novel studies investigating epigenome-wide DNA methylation have offered new opportunities to enhance our understanding of the dynamics within the DNA methylation landscape in CRC. Several studies have characterized DNA methylation profiling and its association with cancer outcomes [8,9], identifying potential tools for diagnosis and prognosis [10,11], disease progression and response to therapy [12]. However, additional research is necessary to introduce DNA methylation into clinical application, both for comprehending the pathogenetic mechanisms and for developing novel therapeutic approaches for CRC. In the present study, we hypothesized that investigating global DNA methylation could provide a more comprehensive insight into the DNA methylation patterns in CRC, potentially revealing new pathogenic mechanisms. Accordingly, we profile genome-wide DNA methylation in CRC by examining tumor and non-adjacent tumor tissues. Our findings highlight *TNXB* (Tenascin XB) as a novel tumor-suppressor gene in CRC, and further functional approaches are analyzed. This study provides evidence supporting a deeper understanding of DNA methylation dysregulation in colorectal carcinogenesis.

## 2. Results

### 2.1. The DNA Methylation Landscape in Colorectal Cancer

We conducted an analysis of epigenome-wide DNA methylation in CRC by comparing tumor and NAT samples. The baseline characteristics of the patients with CRC included in the epigenome-wide DNA methylation are summarized in Appendix A. Principal component analysis (PCA) and heatmap distinctly group the tumor and NAT samples into two well-defined clusters (Figure 1A, Appendix A). Overall, we observed a significant increase in global promoter methylation in the tumor area compared to the NAT area (*p* < 0.001). In contrast, global DNA methylation exhibited an increase in the NAT area in comparison to the tumor area (*p* < 0.001) (Figure 1B). The differential methylation analysis revealed a total of 73,509 DMCs (FDR < 0.001), highlighting the most significantly hypermethylated and hypomethylated positions in genes, including cg21995919 within the *ITGA4* (cg21995919), *OPLAH*, *BTF3* (cg14012294) and *MYBPC3* (cg05649391) genes (Figure 1C, Appendix A; the 20 most significant differentially methylated CpGs are summarized in Appendix A). To find potential genes associated with CRC, we conducted several filtered analysis approaches: (i) with an FDR lower than 1 × 10^−3^, we observed 33,526 hypermethylated and 39,983 hypomethylated DMCs; (ii) through a more stringent analysis identifying DMCs with a LogFC greater than |1.2| and an FDR lower than 1 × 10^−4^, we identified 5124 hypermethylated positions and 3578 hypomethylated positions (Figure 1D). To further evaluate the biological significance of these genes, we conducted Gene Set Enrichment Analysis (GSEA) for both gene ontology (GO) and the KEGG pathway to find pathways activated and suppressed in CRC, related to DNA methylation. The GO-GSEA revealed increases in DNA methylation in several biological pathways, such as RNA and cellular metabolic processes, as well as decreased methylation in extracellular processes (Figure 1E). As for KEGG-GSEA, we found an increase in DNA methylation in the cGMP-dependent protein kinase (PKG) signaling pathway, the TGFβ signaling pathway, cell adhesion molecules, and cAMP signaling pathways, and decreased DNA methylation in amino sugar and nucleotide sugar metabolism (Figure 1F).

Finally, to identify the best candidate gene, we implemented stringent filtering criteria following the workflow outlined in Appendix A, prioritizing LogFC and FDR values, as well as genes exhibiting extended methylation along their loci. This suggests a local and widespread dysregulation of DNA methylation in terms of both the number of DMCs and DMRs (differentially methylated regions; the 20 most significant DMRs are summarized in Appendix A). This analysis led to the identification of the *TNXB* gene as the most promising candidate. Consequently, we focused on a more comprehensive examination of the functionality of the *TNXB* gene in CRC.

### 2.2. TNXB Gene Is Largely Hypomethylated and Overexpressed in Colorectal Cancer

In our exploration of the *TNXB* gene within our cohort, we observed significant hypomethylation of the *TNXB* gene in the tumor area compared to the NAT area (*p* < 0.001). Concurrently, we found decreased mRNA expression in the tumor area in contrast to the NAT area (*p* < 0.01), measured by qPCR and validated by public scRNA-seq data (Figure 2A, Appendix A). To validate these findings, we analyzed the TCGA-COAD and TCGA-READ data, and we found similar results, indicating consistent hypomethylation and decreased mRNA expression of the *TNXB* gene in the tumor area compared to the NAT area (all with *p* < 0.001) (Figure 2B,C). Upon specifically analyzing the total count of DMCs within the *TNXB* gene, in our cohort, we observed a widespread decrease in methylation levels across the gene in the tumor area compared to the NAT area (Figure 2D). Notably, this decrease in methylation was particularly evident within the CpG island, and within DMR 1 and DMR 5 (both with *p* < 0.001) (Appendix A). Intriguingly, DMR 1 encompasses three CpG islands, which demonstrated notable hypomethylation in the tumor area (all with *p* < 0.001). Overall, there was a trend of positive correlation between *TNXB* methylation and expression (r = 0.710; *p* = 0.058) (Figure 2E). Finally, a Kaplan–Meier plot demonstrated that high *TNXB* methylation (the highest tertile) was associated with poorer overall survival compared to low methylation status (the middle and lowest tertiles) (*p* = 0.037) (Figure 2F, adjusted models are represented in the Appendix A). There was also a trend when comparing the median value of *TNXB* methylation in our cohort (*p* = 0.093) with that of TCGA-CAOD (*p* = 0.080) (Appendix A). Finally, we performed a receiver operating characteristic (ROC) curve analysis. The results revealed an area under the curve (AUC) of 0.992 (95% CI: 0.975–1.000), indicating an excellent discriminative capacity (Appendix A).

**Figure 1 ijms-26-07197-f001:**
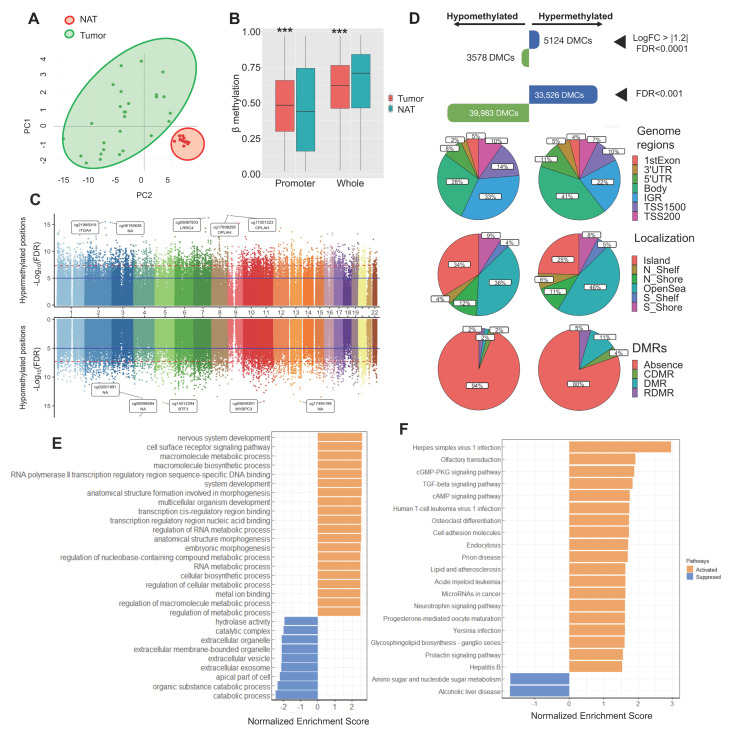
Profile of DNA methylation in tumors from colorectal cancer patients. (**A**) Principal component analysis (PCA) for DNA methylation levels of the 1000 most variable CpGs between the tumor area and NAT area, showing two well-clustered groups. (**B**) Comparison between global DNA methylation and global promoter methylation and the tumor area and the NAT area. Asterisks indicate significant differences between the groups according to the Mann–Whitney test (*** *p* < 0.001). (**C**) Miami plot showing the y axis with the Log_10_(FDR) values of CpGs, and the x axis shows their chromosomal position. This plot displays Log_10_(FDR) for DMC associations for hypermethylated CpG sites in the upper part, and for hypomethylated CpG sites in the lower part. Blue line indicates the genome-wide significance level (*p* < Log_10_(FDR). (**D**) After a stringent analysis with an FDR < 0.001, we found a total of 73,509 DMCs. After selecting by a LogFC > |1.2|, we found 5124 hypermethylated probes and 3578 hypomethylated CpGs. Genomic distribution of the DMCs and their respective locations regarding the broader CpG context over the genome region. (**E**,**F**) Gene Set Enrichment Analysis gene ontology (GO) and Kyoto Encyclopedia of Genes and Genomes (KEGG) pathway analyses, including the most significant DMCs. Abbreviations: DMCs: differentially methylated CpG; FDR: false discovery rate; IGR: intergenic region; NAT: normal adjacent tumor; LogFC: log fold change; TSS: transcription site start; UTR: untranslated region; DMR: differentially methylated regions, (C) cancer (R) reprogramming specific.

**Figure 2 ijms-26-07197-f002:**
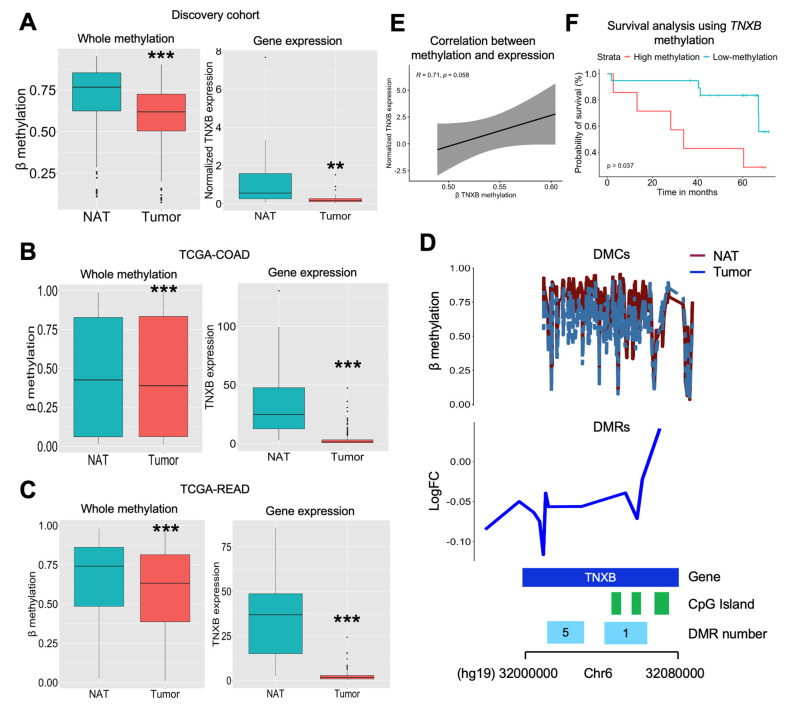
Validation analysis of *TNXB*. (**A**) β methylation of the *TNXB* gene was extracted from the DNA array. We specifically selected DMC sites located at the *TNXB* gene from both tumor and NAT samples. Asterisks indicate significant differences between the groups according to the Mann–Whitney test (** *p* < 0.01, *** *p* < 0.001). Normalized gene expression of *TNXB* in the tumor (N = 30) and NAT (N = 24) areas. Gene expression was normalized using the PPIA gene, and using the following formula: 2^−ΔCt^. Asterisks indicate significant differences between the groups according to the Mann–Whitney test (** *p* < 0.01, *** *p* < 0.001). Gene expression (extracted from RNA-seq analysis) and methylation of the *TNXB* gene extracted from (**B**) TCGA-COAD and (**C**) TCGA-READ, and Asterisks indicate significant differences between the groups according to the Mann–Whitney test (*** *p* < 0.001). (**D**) A representation of the total of DMCs and DMRs located at the *TNXB* gene. Representation of DMR 1 and DMR 5 found in our study and belonging to the *TNXB* gene. (**E**) Correlation analysis using Spearman’s correlation between *TNXB* methylation and expression (N = 8). (**F**) Kaplan–Meier analysis comparing low (lowest and middle tertiles; N = 20) and high (highest tertile; N = 7) methylation analysis.

### 2.3. TNXB Gene Is Found to Be Epigenetically Regulated

To test the effect of DNA methylation on the *TNXB* gene expression, we treated the HCT116 cells with AZA, a demethylating agent, for 72 h. After that, we measured the gene expression of the *TNXB* gene to test the role of DNA methylation in gene expression. Accordingly, we found that AZA treatment induced overexpression of the *TNXB* gene at increasing concentrations (1, 5, and 10 µM) when compared to the vehicle (DMSO) (Figure 3). Interestingly, at 10 µM of AZA, the *TNXB* gene expression was significantly increased when compared to the treatment at 1 µM of AZA.

## 3. Discussion

Our study provides a comprehensive analysis of the DNA methylation profile in tumors from patients with CRC. We observed that colorectal tumors exhibit genome-wide hypomethylation but hypermethylation in the promoter regions. Epigenetically dysregulated genes predominantly influence specific metabolic processes and signaling pathways involved in the regulation of the cell cycle and cell adhesion. Through our filtered analyses, the *TNXB* gene emerges as a potential epimutation in CRC. Notably, *TNXB* is found to be hypomethylated and downregulated within the tumor area. Furthermore, functional analysis reveals that *TNXB* is epigenetically silenced, and downregulation of *TNXB* leads to increased cell migration and proliferation in CRC cell lines, suggesting its role as a tumor-suppressor gene. This identified epigenetic alteration in *TNXB* could have clinical implications, serving as an indicator of malignant potential and the adverse effects associated with disease origin and progression in CRC.

Several studies have confirmed the useful utility of epigenome-wide DNA methylation in cancer research [13]. A recent study conducted by Janssens et al. (2023) identified 13 DMCs when comparing colorectal adenoma vs. colorectal carcinoma, with high discrimination, sensitivity and specificity [14]. Furthermore, a study performed by Baharudin et al. (2022) found a total of 26,093 DMCs, of which 650 were hypermethylated and 25,443 were hypomethylated, when comparing the tumor and NAT areas [9]. Both studies are in line with our findings. This observed phenomenon primarily stems from the loss of methylation in *LINE1*, significantly contributing to the manifestation of aberrant genetic phenotypes. These consequences include chromosomal instability, inactivation of tumor-suppressor genes, and the activation of oncogenes [15]. However, more research is needed to determine whether aberrant DNA methylation occurs early in carcinogenesis or late in the process [16]. In contrast, Naumov et al. (2013) discovered, by comparing differential methylation between cancerous and non-cancerous tumor tissue, more hypermethylated DMCs (10,342 CpG were hypermethylated and 5325 were hypomethylated) [17]. Therefore, more studies are needed to validate the capability of DNA methylation to differentiate between normal and tumor tissues. This distinction represents an ongoing clinical requirement that has yet to be fulfilled.

Overall, several studies were focused on identifying specific mechanisms related to DNA methylation profile to better understand the carcinogenesis process. Several studies have evaluated potential candidates for CRC diagnosis across diverse biological tissues [18,19]. However, their diagnostic accuracy for CRC and effectiveness in early detection in extensive independent cohorts require further enhancement and comprehensive validation [20]. This challenge persists due to the diversity and heterogeneity observed in CRC tumors, alongside additional cancer-independent factors. In some cases, abnormal DNA methylation occurring in a specific locus becomes constitutive. Various epigenetically silenced genes have been identified in CRC, such as *MLH1* or *APC* [21,22]. In our study, we found the aberrant hypomethylation of the *TNXB* gene, in which constitutive silencing may lead to several cancer outcomes, some of which are evaluated in this study.

The *TNXB* gene, responsible for encoding Tenascin-X, plays a crucial role in organizing and preserving the integrity of connective tissues and the extracellular matrix. Its function involves inhibiting cell migration, accelerating collagen fibril formation, and supporting the growth of epithelial tumors and cell plasticity [23]. In the cancer context, *TNXB* expression tends to decrease during cancer progression, while increased expression has been linked to a better prognosis [24]. Despite its significance, there exists a knowledge gap regarding the methylation status of *TNXB* in the existing literature. Barrow and colleagues (2017) discovered significant hypomethylation of four loci associated with the *TNXB* gene in adjacent mucosa from active smokers when compared to never smokers in colorectal tumors [25]. Izquierdo et al. (2021) found a similar pattern in the adipose tissue from patients with CRC when compared to healthy participants [26]. Other studies related *TNXB* to colorectal carcinogenesis, suggesting a potential link [27,28]. In our study, we identified that the *TNXB* gene is largely hypomethylated and downregulated in CRC tumors, which was further validated with the TCGA cohorts, suggesting a potential role in CRC. This hypothesis was confirmed by treating HCT116 cells with a DNA demethylating agent. We found that *TNXB* expression was restored after treatment, indicating that *TNXB* is epigenetically regulated in CRC. Our findings highlight the context-dependent nature of DNA methylation and its effects on gene expression. In our study, most of the differentially methylated CpG sites in *TNXB* were located within the gene body. Gene body methylation has been reported to play diverse regulatory roles, including effects on transcriptional elongation, splicing, or enhancer activity. The downregulation of *TNXB* in tumors despite hypomethylation may thus reflect the influence of additional regulatory mechanisms, such as histone modifications, altered transcription factor binding, or the tumor microenvironment. In contrast, the increased *TNXB* expression observed after AZA treatment in vitro supports a role for DNA methylation in its transcriptional regulation, while also underscoring the simplified nature of in vitro systems compared to primary tumor tissue. Although targeting *TNXB* epigenetic regulation may represent a promising therapeutic avenue, its practical application remains to be explored. Future studies are needed to determine whether restoring *TNXB* expression via demethylating agents or epigenome-editing strategies can produce therapeutic benefits in colorectal cancer models.

This study has several limitations. First, it was conducted in a cohort of patients exclusively from Spain, which may limit the generalizability of the findings to other populations with different genetic backgrounds and environmental exposures. Second, the relatively small sample size may reduce the statistical power to detect subtle methylation differences; thus, validation in larger and independent cohorts is required. Third, although our in vitro experiments suggest a regulatory role of DNA methylation on *TNXB* expression, further functional studies are needed to determine the causal relationship between *TNXB* dysregulation and colorectal cancer pathogenesis. In particular, in vivo experiments using mouse models and advanced 3D culture systems (e.g., organoids) would help to better understand the biological relevance and therapeutic potential of *TNXB* in the context of colorectal cancer. Although the AZA treatment induced overexpression of *TNXB*, this finding should be interpreted with caution. AZA is a global demethylating agent, and its effects are not specific to individual genes. Another limitation of this study is the lack of analysis by stratification based on relevant clinical subtypes such as tumor location (right vs. left colon), TNM stage, and microsatellite instability (MSI/MSS) status. Due to the limited sample size, we were unable to assess whether these factors influence methylation patterns. Finally, this study used adjacent non-tumor tissue (NAT) as the control group, which may not fully represent truly normal colonic mucosa. NAT can exhibit early pre-neoplastic changes or tumor-induced epigenetic alterations, potentially confounding the identification of cancer-specific methylation events.

## 4. Materials and Methods

### 4.1. Study Design and Participants

This study included participants from the University Hospital “Virgen de la Victoria,” recruited between 2012 and 2014. We included 76 patients with CRC who underwent surgery with curative intent. These patients were diagnosed by a pathology specialist using a biopsy and a colonoscopy. All medical records and pathological examinations were reported to confirm the diagnosis. All biopsy samples were classified according to the histological features by pathologists, and to the “World Health Organization Classification of Tumors of the Digestive System” (2016) [29]. Patients with CRC had hemicolectomy and lower anterior resection with ileostomy (due to colorectal carcinoma), followed by total meso-colorectal excision. All patients had at least a 5-year follow-up. This follow-up entailed a clinical visit every three months for the first two years, then every six months beginning in the third year. At each checkup, biochemical variables were measured. Furthermore, a physical examination, biochemical test and a colonoscopy were performed. Finally, we also included 79 healthy participants (cancer free), who underwent hiatus hernia surgery or a cholecystectomy.

Patients with inflammatory acute and chronic diseases, familial polyposis, or those patients who had chemotherapy or radiotherapy treatments were excluded. Participants with infection, renal or cardiovascular diseases, or patients who received treatment that altered their lipid, glucose, calcium, or vitamin D profiles were also excluded.

### 4.2. Samples Included in This Study

Blood samples were obtained from all the participants. Serum samples were extracted from blood samples by centrifugation at 4000 rpm for 15 min at 4 °C. Tumor samples that contain both the tumor area and the normal adjacent tumor-free area (NAT) were obtained from biopsy samples. Formol was used to fix the samples, followed by formalin-fixed paraffin-embedded (FFPE). Pathologists determine the limits between the tumor and adjacent tumor-free area and the percentage of tumoral cells within the tumor area, which is greater than 80% of the tumor cell percentage.

### 4.3. DNA and RNA Extraction

The Qiamp DNA Tissue Kit (Qiagen GmbH, Hilden, Germany) was used to extract genomic DNA. DNA from tumor samples was isolated from 5 to 10 sections of 14 μm of FFPE samples from the tumor area and the NAT area using a Qiamp DNA FFPE Tissue Kit under the instructions of the manufacturer (Qiagen GmbH, Hilden, Germany), with a xylene wash to remove the paraffin. DNA integrity was determined using the NanoDrop ND-1000 (Thermo Fisher Scientific, Indianapolis, IN, USA) at 260/280 and 260/230 ratios and confirmed by electrophoresis in a 1.5% agarose gel in TAE/TBE buffer. The Pico Green dsDNA Quantitation Reagent (Invitrogen, Carlsbad, CA, USA) was used to analyze DNA integrity further and quantify DNA concentrations.

For RNA extraction, RNA from FFPE samples of tumor area and NAT area were isolated from 5 to 10 sections of 14 μm using a RNeasy FFPE Kit under the instructions of the manufacturer (Qiagen GmbH, Hilden, Germany). The Total RNA Purification Kit (Norgen Biotek Corp., Thorold, ON, Canada) was used to isolate total RNA from cells (at a density of 1 × 10^4^ – 1 × 10^5^ cells) according to the manufacturer’s instructions. To generate first-strand cDNA synthesis, we used 1 μg of total extracted RNA, and random primers using the PrimeScript™ RT-PCR Kit as indicated by the manufacturer.

### 4.4. Laboratory Measurements

We measured fasting glucose, total cholesterol, triglycerides, and high-density lipoprotein (HDL) cholesterol levels using the Dimension Autoanalyzer (Dade Behring Inc., Deerfield, IL, USA). We calculated low-density lipoprotein (LDL) cholesterol using the Friedewald equation [30]. Fasting insulin levels were measured using radioimmunoassay methods with BioSource International Inc. (Camarillo, CA, USA). The insulin resistance homeostasis model assessment (HOMA-IR) was calculated using the following equation: HOMA-IR = fasting insulin (IU/mL) fasting glucose (mmol/L)/22.5 [31]. Carcinoembryonic Antigen (CEA) and carbohydrate antigen 19.9 (CA19.9) were measured by ELISA (DRG diagnostics, Marburg, Germany). An ELISA kit (Immundiagnostik, Bensheim, Germany) was used to measure serum 25-hydroxyvitamin D (25(OH)D). 

### 4.5. Bisulfite Reaction and Genome-Wide DNA Methylation Analysis

For the study of genome-wide methylation analysis, high-quality genomic DNA samples (500 ng) from the tumor area (N = 27) and the NAT area (N = 15) were subjected to bisulfite treatment using the EZ-96 DNA Methylation kit (Zymo Research, Irvine, CA, USA) following the manufacturer’s instructions and as previously described [26]. Subsequently, DNA methylation was analyzed by microarray assays using Infinium Human Methylation 450K bead chip technology (Illumina, San Diego, CA, USA). DNA quality checks, bisulfite modification, hybridization, data normalization, statistical filtering, and value calculation were performed. Whole-genome amplification and hybridization were then performed using BeadChip, followed by single-base extension and analysis using the HiScan SQ module (Illumina) to assess the cytosine methylation states. DNA methylation for each CpG site was represented by beta values ranging from 0 to 1, corresponding to fully unmethylated and fully methylated, respectively.

For the validation of global DNA methylation, we used the DNA methylation of *LINE1* (long interspersed nuclear elements 1) as a global DNA methylation marker. Primer sequences are as previously described [4]. A PCR reaction was conducted using a primer concentration of 0.2 nmol/L. DNA pyrosequencing was performed using the PyroMark Q96 ID pyrosequencing System (Qiagen GmbH, Hilden, Germany). The methylation average was presented as the percentage of methylated cytosine over the sum of methylated and unmethylated cytosines. Unmethylated and methylated DNA were used as controls (New England Biolabs, Hertfordshire, UK).

### 4.6. Gene Expression Analysis

For the quantification of *GAPDH* (GlycerAldehyde-3-Phosphate Dehydrogenase) (Hs.PT.39a.22214836) and *TNXB* (Tenascin XB) (Hs.PT.58.39823090.gs) genes, we used commercially available TaqMan primer/probe mix (Integrated DNA Technologies Inc., Madrid, Spain). Gene expression was conducted using Premix Ex Taq^TM^ (Probe qPCR) (*Takara* Bio USA, Inc., Mountain View, CA, USA) according to the instructions of the manufacturer. With TaqMan technology, gene expression was carried out by real-time PCR using QuantStudio 6 Pro (Applied Biosystems, Darmstadt, Germany). Changes in gene expression were normalized by the 2^−ΔCt^ method [32]. The expression results were represented as the target gene/*PPIA* ratio.

### 4.7. Cell Culture

HCT116 (ATCC CCL-247^™^) and Caco2 (ATCC HTB-37^™^) human colorectal carcinoma cells were cultured in Dulbecco′s Modified Eagle′s Medium—low glucose (Biowest, Nuaillé, France), plus 10% fetal bovine serum (FBS) (Gibco, Carlsbad, CA, USA), 1% of L-glutamine and 1% of streptomycin/penicillin at 37 °C and 5% CO_2_. Cells were periodically checked for mycoplasma using DAPI analysis [33] and PCR using universal primers [34]. HCT116 are maintained at passages lower than 20. All cell experiments were conducted in triplicates and three independent replicates.

### 4.8. DNA Demethylation *In Vitro*

For 5-Aza-2-deoxycytidine (AZA) (A3656, Sigma Aldrich, Madrid, Spain) treatment as an inhibitor of DNA methyltransferases (DNMTs), cells were treated with 1 µM, 5 µM and 10 µM of AZA for 72 h. The media were renewed every 48 h. RNA extraction was performed after 72 h of treatment.

### 4.9. Bioinformatic Analysis: DNA Methylation Analysis, TCGA Data and Single-Cell Analysis

The minfi package and the ChAMP pipeline from Bioconductor were used with default settings for DNA methylation analysis [35,36]. Probes with *p*-values above 0.01 were excluded. Normalization and preprocessing were conducted using the minfi package. Probes were stratified for quantile normalization preprocessing for Illumina methylation microarrays. Probes were stratified by region, and bad samples were eliminated using the median of methylated and unmethylated signal for each sample. Probes with single-nucleotide polymorphisms at CpG or single-base extension sites and sex chromosomes were removed. Finally, we obtained M values for further investigation. For the ChAMP pipeline, quality checks and normalization of β values were performed using the BMIQ method. Principal component analysis (PCA) and total and promoter methylation were conducted using the ChAMP pipeline. The Miami plot was conducted using the qqman package [37]. Differential analysis was adjusted for age, sex and BMI (obesity was classified with a BMI ≤ 25 kg/m^2^). The gene–protein interaction network was constructed according to STRING analysis [38]. Gene ontology (GO) and Kyoto Encyclopedia of Genes and Genomes (KEGG) pathway analysis were conducted to explore the function of the genes related to CRC using the gometh function in the missMethyl package [37].

TCGA-COAD (colon adenocarcinoma) and TCGA-READ (rectal adenocarcinoma) methylation data were obtained from The Cancer Genome Atlas (TCGA) database (Cancer Genome Atlas Research) [39]. IDAT files from 308 cancer cases and 41 NAT from TCGA-COAD and 94 cancer cases and 10 NAT from TCGA-READ were included, downloaded and processed using the TCGAbiolinks R packages [40]. Data were filtered, normalized and preprocessed following similar protocols as described above. Survival analyses were extracted from the OncoDB web tool (http://oncodb.org) [41]. We used normalized RNA-sequencing (RNA-seq) expression data from TCGA-COAD and TCGA-READ using the TCGAbiolinks R package to identify differentially expressed genes. Results from single-cell analysis using 52,609 cells were extracted from results published by Lee, H.O. et al. (2020) [42] using the EMBL-EBI platform: https://www.ebi.ac.uk/.

### 4.10. Statistical Analysis

The results are presented as the mean ± standard deviation (SD) for continuous variables and as numbers (percentages) for categorical variables. Student’s *t*-test or the Mann–Whitney test was applied according to the normality of the variables. Pearson correlation coefficients between methylation and anthropometric and biochemical parameters and multivariate linear regression were performed. Kaplan–Meier curves were used for overall survival analyses. The hazard ratio (HR) was performed using multivariate Cox proportional hazard regression for methylation. To assess the diagnostic performance of *TNXB* methylation in differentiating tumor tissue from adjacent non-tumor tissue (NAT), we conducted a receiver operating characteristic (ROC) curve analysis using the pROC R package. Methylation levels were compared between tumor and NAT samples, and the area under the curve (AUC) was calculated with its 95% confidence interval using the DeLong method. Analyses and graphic representation were pointed out, performed using R v.3.5.1 software (Integrated Development for R. RStudio, PBC, Boston, MA, USA), and the significance *p* value was set at *p* < 0.05 [43].

## 5. Conclusions

Our study, employing epigenome-wide DNA methylation analysis, underscores dysregulation in the epigenetic landscape in CRC. We found overall hypomethylation in tumor tissue, primarily focusing on the *TNXB* gene. *TNXB* was mainly hypomethylated and downregulated. Functional analysis revealed that *TNXB* is epigenetically regulated. Further research is necessary to gain a deeper understanding of the role of *TNXB* in the pathophysiology of CRC.

## Figures and Tables

**Figure 3 ijms-26-07197-f003:**
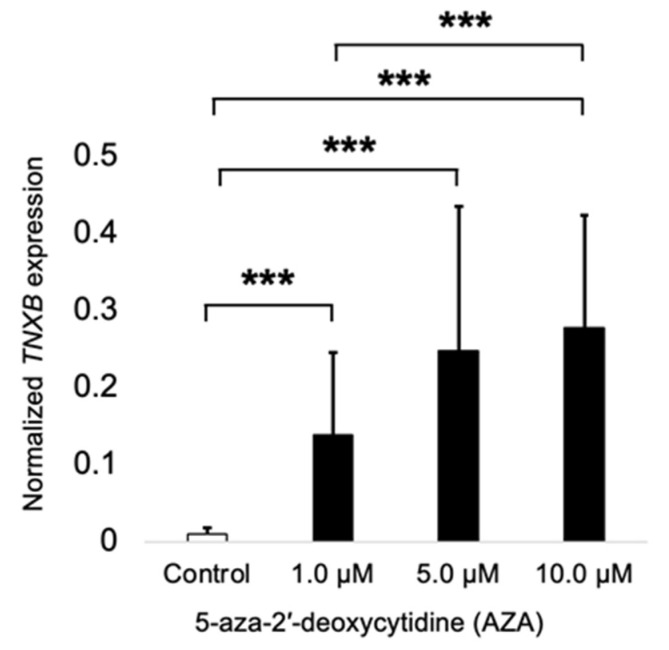
Epigenetic dysregulation of *TNXB* in colorectal cancer cell line HCT116. HCT116 cells were treated with 5-Aza-2′-deoxycytidine (AZA) for 72 h to induce DNA demethylation. *TNXB* gene expression was measured by quantitative RT-PCR. The results indicate that epigenetic silencing via DNA methylation regulates *TNXB* expression in colorectal cancer cells. Abbreviations: AZA, 5-Aza-2′-deoxycytidine; *TNXB*, Tenascin XB. *** *p* < 0.001.

## Data Availability

Data is contained within the article or Appendix A.

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
