# Peer review of "Unraveling TNXB Epigenetic Alterations Through Genome-Wide DNA Methylation Analysis and Their Implications for Colorectal Cancer"

_ijms, 2025, doi:10.3390/ijms26157197_

Round 1

Reviewer 1 Report

Comments and Suggestions for Authors
  1. This study analyzed genome-wide DNA methylation in only 27 tumor samples and 15 adjacent normal tissues (NAT), which limits statistical power and fails to adequately capture the heterogeneity of colorectal cancer (CRC).
  2. The study lacks stratification based on relevant clinical subtypes, such as tumor location (right vs. left colon), disease stage (TNM), and microsatellite instability status (MSI/MSS), all of which may significantly impact methylation patterns and introduce confounding bias.
  3. The study uses adjacent non-tumor tissue (NAT) as the control group. However, NAT may already exhibit pre-neoplastic changes or tumor-induced microenvironmental alterations, making it an imperfect "normal" reference.
  4. Although the authors apply thresholds such as FDR < 0.001 and LogFC > |1.2|, they do not report whether cross-validation or independent dataset testing was performed. There is also no evidence of statistical robustness metrics such as ROC curves or AUC for the identified DMCs/DMRs, and the study does not discuss batch effect correction or sample randomization, especially critical when using FFPE samples prone to degradation artifacts.
  5. While TNXB hypomethylation and downregulation are both observed in tumors, the correlation (r = 0.710, p = 0.058) is marginally significant and insufficient to claim direct epigenetic regulation. Furthermore, the authors do not clarify the location of methylation changes (e.g., TSS200, TSS1500, shores, shelves), which is essential for interpreting their regulatory impact on transcription.
  6. Although TNXB is proposed as a tumor suppressor based on its hypomethylation and downregulation, the study lacks direct functional experiments such as knockdown or overexpression assays to validate its tumor-suppressive effects.
  7. The AZA demethylation assay provides indirect support, but AZA is a broad-spectrum demethylating agent, and TNXB-specific effects cannot be inferred.
  8. The Kaplan–Meier survival analysis uses a small sample size and tertile-based stratification without adjusting for potential confounders (e.g., age, sex, tumor stage).
  9. Although the study suggests TNXB methylation as a potential biomarker, no ROC curve, sensitivity, specificity, or predictive value assessments were conducted. The biomarker's performance was not compared against existing methylation-based CRC markers (e.g., SFRP2, SEPT9), limiting its relevance in clinical screening.
  10. The study does not examine correlations between TNXB methylation and clinical outcomes such as response to chemotherapy, CEA/CA19-9 levels, or recurrence, and its utility in clinical stratification, monitoring, or treatment decision-making remains unclear.
Comments on the Quality of English Language

The English could be improved to more clearly express the research.

Author Response

Reviewer 1

  1. This study analyzed genome-wide DNA methylation in only 27 tumor samples and 15 adjacent normal tissues (NAT), which limits statistical power and fails to adequately capture the heterogeneity of colorectal cancer (CRC).
  2. The study lacks stratification based on relevant clinical subtypes, such as tumor location (right vs. left colon), disease stage (TNM), and microsatellite instability status (MSI/MSS), all of which may significantly impact methylation patterns and introduce confounding bias.
  3. The study uses adjacent non-tumor tissue (NAT) as the control group. However, NAT may already exhibit pre-neoplastic changes or tumor-induced microenvironmental alterations, making it an imperfect "normal" reference.
  4. Although the authors apply thresholds such as FDR < 0.001 and LogFC > |1.2|, they do not report whether cross-validation or independent dataset testing was performed. There is also no evidence of statistical robustness metrics such as ROC curves or AUC for the identified DMCs/DMRs, and the study does not discuss batch effect correction or sample randomization, especially critical when using FFPE samples prone to degradation artifacts.
  5. While TNXB hypomethylation and downregulation are both observed in tumors, the correlation (r = 0.710, p = 0.058) is marginally significant and insufficient to claim direct epigenetic regulation. Furthermore, the authors do not clarify the location of methylation changes (e.g., TSS200, TSS1500, shores, shelves), which is essential for interpreting their regulatory impact on transcription.
  6. Although TNXB is proposed as a tumor suppressor based on its hypomethylation and downregulation, the study lacks direct functional experiments such as knockdown or overexpression assays to validate its tumor-suppressive effects.
  7. The AZA demethylation assay provides indirect support, but AZA is a broad-spectrum demethylating agent, and TNXB-specific effects cannot be inferred.
  8. The Kaplan–Meier survival analysis uses a small sample size and tertile-based stratification without adjusting for potential confounders (e.g., age, sex, tumor stage).
  9. Although the study suggests TNXB methylation as a potential biomarker, no ROC curve, sensitivity, specificity, or predictive value assessments were conducted. The biomarker's performance was not compared against existing methylation-based CRC markers (e.g., SFRP2, SEPT9), limiting its relevance in clinical screening.
  10. The study does not examine correlations between TNXB methylation and clinical outcomes such as response to chemotherapy, CEA/CA19-9 levels, or recurrence, and its utility in clinical stratification, monitoring, or treatment decision-making remains unclear.

Response

Thank you to the reviewer for their valuable comments. Please find point-by-point our response to the reviewer’s concern, with the objective to improve the quality of our manuscript.

Point 1

This study analyzed genome-wide DNA methylation in only 27 tumor samples and 15 adjacent normal tissues (NAT), which limits statistical power and fails to adequately capture the heterogeneity of colorectal cancer (CRC).

Response 1.

We appreciate the comment of the reviewer and fully agree that the limited sample size is an important consideration.

As acknowledged in the Limitation section (already added) of the manuscript, the relatively small cohort may reduce statistical power and may not fully reflect the molecular heterogeneity of colorectal cancer. To address this, we applied stringent statistical thresholds and multiple filtering strategies to minimize false positives. Additionally, our main findings—particularly the hypomethylation and downregulation of TNXB—were cross-validated using publicly available datasets, including TCGA, supporting the robustness of our observations. Nevertheless, we recognize the need for further validation in larger and more diverse cohorts and have emphasized this in the revised manuscript in the Limitation section.

Point 2

The study lacks stratification based on relevant clinical subtypes, such as tumor location (right vs. left colon), disease stage (TNM), and microsatellite instability status (MSI/MSS), all of which may significantly impact methylation patterns and introduce confounding bias.

Response 2

We thank the reviewer for this insightful comment.

We fully acknowledge that clinical subtypes such as tumor location, TNM stage, and MSI/MSS status may influence DNA methylation profiles and introduce heterogeneity. Unfortunately, some of these variables were either incomplete or unavailable for a subset of our cohort, (but Supplementary Table 1 included data about TNM stage, tumor location and survival rate) limiting our ability to perform stratified analyses with sufficient statistical power. Nonetheless, we agree this is an important aspect, and we have added a statement in the Limitations section to highlight the need for future studies with detailed clinical annotations to explore subtype-specific methylation patterns.

Another limitation of this study is the lack of analysis by stratification based on relevant clinical subtypes such as tumor location (right vs. left colon), TNM stage, and microsatellite instability (MSI/MSS) status. Due to the limited sample size, we were unable to assess whether these factors influence methylation patterns.

Point 3

The study uses adjacent non-tumor tissue (NAT) as the control group. However, NAT may already exhibit pre-neoplastic changes or tumor-induced microenvironmental alterations, making it an imperfect "normal" reference.

Response 3

We thank the reviewer for raising this important point.

We agree that adjacent non-tumor tissue (NAT) may not fully represent healthy normal tissue, as it can harbor early molecular alterations or microenvironmental changes influenced by the tumor. We chose NAT due to its matched origin and availability, which reduces inter-individual variability. Nevertheless, we have now acknowledged this limitation in the Limitations section, and we recognize the importance of including truly normal mucosa from healthy individuals in future studies:

“This study used adjacent non-tumor tissue (NAT) as the control group, which may not fully represent truly normal colonic mucosa. NAT can exhibit early pre-neoplastic changes or tumor-induced epigenetic alterations, potentially confounding the identification of cancer-specific methylation events.”

Point 4

Although the authors apply thresholds such as FDR < 0.001 and LogFC > |1.2|, they do not report whether cross-validation or independent dataset testing was performed. There is also no evidence of statistical robustness metrics such as ROC curves or AUC for the identified DMCs/DMRs, and the study does not discuss batch effect correction or sample randomization, especially critical when using FFPE samples prone to degradation artifacts.

Response 4

Thank you to the reviewer for this recommendation.

In response, we performed a ROC curve analysis using the methylation levels of TNXB in tumor and NAT samples. The analysis yielded an AUC of 0.992 (95% CI: 0.975–1.000), indicating excellent discriminatory power. The optimal cut-off point for methylation was 0.611, which provided a sensitivity of 96.3% and a specificity of 100%. These results support the robustness and potential clinical relevance of TNXB hypomethylation as a biomarker in colorectal cancer. We have now included these results in the revised manuscript (Results section and Supplementary Figure 2E).

Point 5

While TNXB hypomethylation and downregulation are both observed in tumors, the correlation (r = 0.710, p = 0.058) is marginally significant and insufficient to claim direct epigenetic regulation. Furthermore, the authors do not clarify the location of methylation changes (e.g., TSS200, TSS1500, shores, shelves), which is essential for interpreting their regulatory impact on transcription.

Response 5

We thank the reviewer for this thoughtful comment.

As noted, the correlation between TNXB methylation and expression was moderate, possibly due to the limited sample size. Nevertheless, we believe the observed trend is biologically relevant and warrants further investigation in larger cohorts. Importantly, methylation values were calculated as the average β-value across all CpG probes annotated to the TNXB gene. As shown in Figure 2D, the majority of these CpGs are located within the gene body, rather than in promoter regions such as TSS200 or TSS1500. This distribution suggests that TNXB may be subject to non-canonical regulatory mechanisms, as gene body methylation has been increasingly recognized for its role in modulating transcription elongation, splicing, or even acting independently of expression changes.

We have clarified this point in the revised version of the manuscript in the Discussion section and included further discussion regarding the functional interpretation of gene body methylation in the context of TNXB regulation.

Point 6

Although TNXB is proposed as a tumor suppressor based on its hypomethylation and downregulation, the study lacks direct functional experiments such as knockdown or overexpression assays to validate its tumor-suppressive effects.

Response 6

We thank the reviewer for this insightful comment.

We agree that functional validation is essential to confirm the tumor-suppressive role of TNXB. In this study, we focused on epigenomic and transcriptomic analyses, supported by data from patient samples and public databases. While our findings are suggestive of a potential tumor-suppressive role of TNXB in CRC, we acknowledge the lack of direct functional assays. We have now included this point in the Limitations sectun. Future studies will aim to address this important aspect.

Point 7

The AZA demethylation assay provides indirect support, but AZA is a broad-spectrum demethylating agent, and TNXB-specific effects cannot be inferred.

Response 7

Thank you to the reviewer for this insight.

We acknowledge that AZAis a broad-spectrum DNA demethylating agent, and its effects are not gene-specific. The overexpression of TNXB observed following AZA treatment provides indirect support for the role of DNA methylation in regulating TNXB expression. However, we fully agree that TNXB-specific effects cannot be definitively attributed to AZA treatment alone.

To address this limitation, we have added a statement in the Discussion acknowledging the nonspecific nature of AZA and the need for future functional assays, such as targeted TNXB promoter demethylation or CRISPR/dCas9-based epigenetic editing, to confirm a direct causal relationship.

“Although the AZA treatment induced overexpression of TNXB, this finding should be interpreted with caution. AZA is a global demethylating agent, and its effects are not specific to individual genes”

Point 8

The Kaplan–Meier survival analysis uses a small sample size and tertile-based stratification without adjusting for potential confounders (e.g., age, sex, tumor stage).

Response 8

Thank you for your insightful comment regarding potential confounders in the survival analysis.

We have adjusted the Cox proportional hazards model for age and tumor location, as well as additionally including sex. The adjusted results show that high TNXB methylation remains associated with poorer survival, with a hazard ratio (HR) of 6.74 (95% CI 1.01–44.80, p = 0.048) when adjusting for age and tumor location. When sex is also included in the model, the HR increases to 9.26 (95% CI 0.82–104.83, p = 0.072), showing a trend toward significance despite the limited sample size. These analyses have been added to Supplementary Figure 2C to provide a clearer understanding of the adjusted survival associations.

Point 9

Although the study suggests TNXB methylation as a potential biomarker, no ROC curve, sensitivity, specificity, or predictive value assessments were conducted. The biomarker's performance was not compared against existing methylation-based CRC markers (e.g., SFRP2, SEPT9), limiting its relevance in clinical screening.

Response 9

Thank you to the reviewer for this comment.

This point was already answered in the previous comments.

Although we acknowledge that we have not directly compared TNXB against other markers such as SFRP2 or SEPT9, our findings highlight its strong potential as a candidate biomarker and warrant further comparative studies in larger, independent cohorts.

Point 10

The study does not examine correlations between TNXB methylation and clinical outcomes such as response to chemotherapy, CEA/CA19-9 levels, or recurrence, and its utility in clinical stratification, monitoring, or treatment decision-making remains unclear.

Response 10

We really appreciate the valuable observation of the reviewer.

This study was designed as a genome-wide screening effort aimed at identifying differentially methylated genes in colorectal cancer. Given the limited sample size and the lack of detailed longitudinal clinical data, assessing associations between TNXB methylation and specific clinical outcomes such as treatment response, serum biomarkers (CEA/CA19-9), or recurrence risk was beyond the scope of this work. We agree that such correlations are of great clinical importance, but they typically require larger and more extensively annotated patient cohorts, such as those used in prospective clinical studies.

Nonetheless, we believe our data provide solid preliminary evidence supporting TNXB as a potential biomarker, based on robust differential methylation, expression analyses, and ROC curve performance. We have now acknowledged this limitation in the Discussion and emphasized that future studies will focus on precisely these aspects to fully explore the translational potential of TNXB in clinical stratification and monitoring.

Reviewer 2 Report

Comments and Suggestions for Authors

This paper reports a study on epigenetic changes and DNA methylation analysis of the TNXB gene in the pathogenesis of colorectal cancer. The TNXB gene plays a vital role in the structure and maintenance of connective tissue and the extracellular matrix. Tenascin-X, encoded by this gene, functions to inhibit cell migration, promote collagen fiber formation, and support the growth and plasticity of epithelial tumors. According to the research, TNXB expression tends to decrease during cancer progression, and increased expression is associated with a favorable prognosis. In this study, it was confirmed that the TNXB gene is mainly hypomethylated and its expression is reduced in colorectal cancer (CRC). Furthermore, decreased TNXB appears to promote cell migration and proliferation, suggesting its possible role as a tumor suppressor gene. The epigenetic regulation of this gene is believed to play a crucial role in the pathophysiology of CRC. This is a fascinating study. However, several concerns are noted:

  1. In the introduction, for readers unfamiliar with the topic, I recommend including a definition of epigenetics and, if possible, a figure illustrating the genetic and epigenetic changes associated with colorectal cancer.
  2. Regarding DNA methylation, please provide a brief overview of its discovery and the mechanisms underlying abnormal DNA methylation in human cancer.
  3. While the impact of TNXB hypomethylation on colorectal cancer is described in detail, the influence of external factors such as diet, lifestyle, and environmental factors on methylation status is not addressed. Additionally, since the study population is limited to patients from Spain, it is unclear whether the results are reproducible in different racial or regional populations. Please discuss specifically the practicality of epigenetic regulation of the TNXB gene as a therapeutic target and the feasibility of its application.
  4. Please check the structure of the article (e.g., material and methods and the discussion)

Comments on the Quality of English Language

Minor English editing is needed.

Author Response

Reviewer 2

This paper reports a study on epigenetic changes and DNA methylation analysis of the TNXB gene in the pathogenesis of colorectal cancer. The TNXB gene plays a vital role in the structure and maintenance of connective tissue and the extracellular matrix. Tenascin-X, encoded by this gene, functions to inhibit cell migration, promote collagen fiber formation, and support the growth and plasticity of epithelial tumors. According to the research, TNXB expression tends to decrease during cancer progression, and increased expression is associated with a favorable prognosis. In this study, it was confirmed that the TNXB gene is mainly hypomethylated and its expression is reduced in colorectal cancer (CRC). Furthermore, decreased TNXB appears to promote cell migration and proliferation, suggesting its possible role as a tumor suppressor gene. The epigenetic regulation of this gene is believed to play a crucial role in the pathophysiology of CRC. This is a fascinating study. However, several concerns are noted:

  1. In the introduction, for readers unfamiliar with the topic, I recommend including a definition of epigenetics and, if possible, a figure illustrating the genetic and epigenetic changes associated with colorectal cancer.
  2. Regarding DNA methylation, please provide a brief overview of its discovery and the mechanisms underlying abnormal DNA methylation in human cancer.
  3. While the impact of TNXB hypomethylation on colorectal cancer is described in detail, the influence of external factors such as diet, lifestyle, and environmental factors on methylation status is not addressed. Additionally, since the study population is limited to patients from Spain, it is unclear whether the results are reproducible in different racial or regional populations. Please discuss specifically the practicality of epigenetic regulation of the TNXB gene as a therapeutic target and the feasibility of its application.
  4. Please check the structure of the article (e.g., material and methods and the discussion)

Response

Thank you to the reviewer for their valuable comments. Please find point-by-point our response to the reviewer’s concern, with the objective to improve the quality of our manuscript.

Point 1

In the introduction, for readers unfamiliar with the topic, I recommend including a definition of epigenetics and, if possible, a figure illustrating the genetic and epigenetic changes associated with colorectal cancer.

Response 1.

Thank you to the reviewer for this comment.

We thank the reviewer for this helpful suggestion. To improve clarity for readers less familiar with the topic, we have now included a brief definition of epigenetics in the Introduction. Please, see the following text introduced in the Introduction section:

“Epigenetics refers to heritable changes in gene expression without alterations in the DNA sequence. These modifications mainly include DNA methylation and histone modifications, which regulate gene accessibility. In CRC, epigenetic alterations, such as aberrant DNA methylation play a key role in tumor initiation and progression”

Point 2

Regarding DNA methylation, please provide a brief overview of its discovery and the mechanisms underlying abnormal DNA methylation in human cancer.

Response 2.

Thank you to the reviewer for this suggestion.

In fact, this recommendation was already addressed in the first draft of the manuscript. In the Introduction section, we included a paragraph explaining the role of DNA methylation in human tumors, highlighting its implication in colorectal cancer pathogenesis:

“This hypomethylation potentially plays a critical role in promoting carcinogenesis by modulating genomic instability, poorer prognoses and response to chemotherapy [6]. Alterations in DNA methylation have been shown to be useful for clinical application, since they may occur early in oncogenesis, are stable, reversible, and can be assayed in many tissues by no-invasive methods [7].”

Point 3.

While the impact of TNXB hypomethylation on colorectal cancer is described in detail, the influence of external factors such as diet, lifestyle, and environmental factors on methylation status is not addressed. Additionally, since the study population is limited to patients from Spain, it is unclear whether the results are reproducible in different racial or regional populations. Please discuss specifically the practicality of epigenetic regulation of the TNXB gene as a therapeutic target and the feasibility of its application.

Response 3.

Thank you to the reviewer for this valuable observation.

We understand that DNA methylation can be influenced by several external factors. To address this, our differential methylation analysis was already adjusted for potential confounders, including age, sex, and BMI (obesity was classified as BMI ≥ 25 kg/m²). This statement is clarified in the Method Section.

Additionally, we have added the following statement in the Limitations section regarding the study population being limited to patients from Spain:

This study has several limitations. First, it was conducted in a cohort of patients exclusively from Spain, which may limit the generalizability of the findings to other populations with different genetic backgrounds and environmental exposures. Second, the relatively small sample size may reduce the statistical power to detect subtle methylation differences; thus, validation in larger and independent cohorts is required. Third, although our in vitro experiments suggest a regulatory role of DNA methylation on TNXB expression, further functional studies are needed to determine the causal relationship between TNXB dysregulation and colorectal cancer pathogenesis. In particular, in vivo experiments using mouse models and advanced 3D culture systems (e.g., organoids) would help to better understand the biological relevance and therapeutic potential of TNXB in the context of colorectal cancer

Regarding the comment on the therapeutic potential of epigenetic regulation of the TNXB gene and the feasibility of its application, we have added the following statement:

“Although targeting TNXB epigenetic regulation may represent a promising therapeutic avenue, its practical application remains to be explored. Future studies are needed to determine whether restoring TNXB expression via demethylating agents or epigenome-editing strategies can produce therapeutic benefits in colorectal cancer models.”

Point 4

Please check the structure of the article (e.g., material and methods and the discussion)

Response 4.

We thank the reviewer for this observation.

The current structure of the manuscript, in which the Materials and Methods section is placed after the Discussion, follows the formatting guidelines recommended by the target journal. We have verified that all sections are correctly ordered according to the journal’s instructions for authors.

Reviewer 3 Report

Comments and Suggestions for Authors

The authors compared 27 tumors and 15 normal tissues and revealed 78,935 differentially methylated CpG sites. They claim that TNXB gene is hypomethylated and downregulated in colorectal cancer. This can be a chance finding due multiple comparisons. There is a disagreement between the  results obtained in tumor samples and cell line experiments: “…we observed significant hypomethylation of the TNXB gene in the tumor area compared to the NAT area. Concurrently, we found a decreased mRNA expression in the tumor area in contrast to the NAT area”. “Accordingly, we found that AZA treatment [i.e., demethylating treatment] induced overexpression of the TNXB gene at crescent concentrations”. There is a missing word in the title: “Unraveling TNXB Epigenetic Alterations through Genome wide DNA methylation [Аnalysis] and their Implications for Colorectal Cancer Pathogenesis”.

Author Response

Reviewer 3

The authors compared 27 tumors and 15 normal tissues and revealed 78,935 differentially methylated CpG sites. They claim that TNXB gene is hypomethylated and downregulated in colorectal cancer. This can be a chance finding due multiple comparisons. There is a disagreement between the  results obtained in tumor samples and cell line experiments: “…we observed significant hypomethylation of the TNXB gene in the tumor area compared to the NAT area. Concurrently, we found a decreased mRNA expression in the tumor area in contrast to the NAT area”. “Accordingly, we found that AZA treatment [i.e., demethylating treatment] induced overexpression of the TNXB gene at crescent concentrations”. There is a missing word in the title: “Unraveling TNXB Epigenetic Alterations through Genome wide DNA methylation [Аnalysis] and their Implications for Colorectal Cancer Pathogenesis”.

Response

Thank you to the reviewer for their valuable comments. Please find point-by-point our response to the reviewer’s concern, with the objective to improve the quality of our manuscript.

Point 1

The authors compared 27 tumors and 15 normal tissues and revealed 78,935 differentially methylated CpG sites. They claim that TNXB gene is hypomethylated and downregulated in colorectal cancer. This can be a chance finding due multiple comparisons.

Response 1

We appreciate this concern from the reviewer regarding the possibility of chance findings due to multiple comparisons. To mitigate this, a detailed stringent filtering was explained in the Result Section. First, we used a false discovery rate (FDR) threshold of 1×10⁻³ to initially identify 33,526 hypermethylated and 39,983 hypomethylated CpG sites. We then applied a more stringent threshold (FDR < 1×10⁻⁴ and |LogFC| > 1.2), resulting in a refined set of 8,702 DMCs (5,124 hypermethylated and 3,578 hypomethylated sites).

To further prioritize biologically relevant findings, we performed Gene Set Enrichment Analyses (GSEA) based on both GO and KEGG annotations to identify pathways significantly associated with DNA methylation changes in colorectal cancer, revealing process such as extracellular matrix, in which TNXB is a key player. Finally, we applied a stepwise filtering pipeline (outlined in Supplementary Figure 1B), integrating statistical stringency (FDR and effect size), number of DMCs and DMRs, and spatial methylation patterns across gene loci. This multi-step strategy led us to identify TNXB as a gene exhibiting extended and significant hypomethylation, along with consistent downregulation in CRC, suggesting a functional and potentially non-random association.

Overall, this pipeline is not based on a single CpG site or result, but is supported by a robust and systematic analytical approach designed to minimize the likelihood of false positives.

Point 2

There is a disagreement between the  results obtained in tumor samples and cell line experiments: “…we observed significant hypomethylation of the TNXB gene in the tumor area compared to the NAT area. Concurrently, we found a decreased mRNA expression in the tumor area in contrast to the NAT area”. “Accordingly, we found that AZA treatment [i.e., demethylating treatment] induced overexpression of the TNXB gene at crescent concentrations”.

Response 2

Thank you to the reviewer for this observation.

TNXB shows hypomethylation in tumor tissues and is downregulated, a finding that is also supported by TCGA data. Although not shown in the manuscript, TCGA data revealed a predominantly negative correlation between DNA methylation and TNXB expression. This observation is intriguing and suggests a complex regulatory landscape. Importantly, most of the differentially methylated CpG sites in TNXB found in our manuscript are located within the gene body. Methylation here is known to have context-dependent effects. In some cases, gene body hypomethylation can be associated with transcriptional repression, particularly when it affects splicing or enhancer activity.

AZA-induced demethylation in cell lines leads to increased TNXB expression. This observation reflects the complexity of epigenetic regulation in primary tumors compared to in vitro models. Overall, AZA treatment causes global demethylation, including promoter and regulatory regions, which may release transcriptional repression and result in increased TNXB expression in vitro. This supports a role for DNA methylation in modulating TNXB expression, but also highlights those other regulatory layers (e.g., histone modifications, transcription factors, tumor microenvironment) likely contribute to its downregulation in colorectal tumors.

We have clarified this point in the Discussion section to reflect the context-dependent nature of DNA methylation and gene expression.

“Our findings highlight the context-dependent nature of DNA methylation and its effects on gene expression. In our study, most of the differentially methylated CpG sites in TNXB were located within the gene body. Gene body methylation has been reported to play diverse regulatory roles, including effects on transcriptional elongation, splicing, or enhancer activity. The downregulation of TNXB in tumors despite hypomethylation may thus reflect the influence of additional regulatory mechanisms, such as histone modifications, altered transcription factor binding, or the tumor microenvironment. In contrast, the increased TNXB expression observed after AZA treatment in vitro supports a role for DNA methylation in its transcriptional regulation, while also underscoring the simplified nature of in vitro systems compared to primary tumor tissue.”

Point 3

There is a missing word in the title: “Unraveling TNXB Epigenetic Alterations through Genome wide DNA methylation [Аnalysis] and their Implications for Colorectal Cancer Pathogenesis”.

Response 3

Thank you for your observation. We have corrected the title to read: “Unraveling TNXB Epigenetic Alterations through Genome-wide DNA Methylation Analysis and their Implications for Colorectal Cancer.”

Round 2

Reviewer 1 Report

Comments and Suggestions for Authors

The author has revised the manuscript according to suggestion.

Reviewer 2 Report

Comments and Suggestions for Authors

The authors have revised their manuscript, and it has been much improved. 

Reviewer 3 Report

Comments and Suggestions for Authors

I apologize, I am not convinced by arguments raised by the authors.